# Interleukin-6 and Lymphocyte-to-Monocyte Ratio Indices Identify Patients with Intrahepatic Cholangiocarcinoma

**DOI:** 10.3390/biomedicines12040844

**Published:** 2024-04-11

**Authors:** Teerachat Saeheng, Juntra Karbwang, Kesara Na-Bangchang

**Affiliations:** 1Center of Excellence in Pharmacology and Molecular Biology of Malaria and Cholangiocarcinoma, Chulabhorn International College of Medicine, Thammasat University (Rangsit Campus), 99, moo 18, Phaholyothin Road, Klongneung Sub-District, Klongluang District, Pathum Thani 12121, Thailand; itachi_bmed@hotmail.com; 2Drug Discovery and Development Center, Office of Advanced Science and Technology, Thammasat University (Rangsit Campus), Pathum Thani 12121, Thailand; jkarbwang@yahoo.com

**Keywords:** biomarkers, prognostic predictor, intrahepatic cholangiocarcinoma, biliary tract cancer, hepatobiliary cancer

## Abstract

Background and aims: Intrahepatic cholangiocarcinoma (iCCA) is a fatal biliary tract cancer with a dismal prognosis due to ineffective diagnostic tools with limited clinical utility. This study investigated peripheral blood indices and cytokine levels to diagnose iCCA. Methods: Blood samples were collected from healthy subjects (*n* = 48) and patients with advanced-stage iCCA (*n* = 47) during a phase I and then phase II trial, respectively. Serum cytokines were measured using a flow cytometer. The peripheral blood indices were estimated based on laboratory data. Multi-linear regression analysis was applied, followed by a probability transformation. The cut-off value and model accuracy were determined using the receiver operating curve (ROC) and the area under the curve (AUC). Results: The interleukin-6 (IL6) and lymphocyte-to-monocyte ratio (LMR) were potential predictors of iCCA [AUC = 0.91 (0.85–0.97) and 0.81 (0.68–0.93); sensitivity = 0.70 and 0.91; specificity = 0.91 and 0.85, respectively]. Patients with IL6 concentrations higher than 11.635 pg/mL (OR = 23.33, *p* < 0.001) or LMR lower than 7.2 (OR = 58.08, *p* < 0.001) are at risk of iCCA development. Patients with IL6 levels higher than 21.83 pg/mL, between 15.95 and 21.83 pg/mL, between 8.8 and 15.94 pg/mL, and lower than 8.8 pg/mL were classified as very high-, high-, intermediate-, and low-risk, respectively. Patients with an LMR between 1 and 3.37, 3.38 and 5.76, 5.77 and 7.18, and higher than 7.18 were classified as very high-, high-, intermediate-, and low-risk, respectively. Conclusions: LMR is recommended for iCCA screening since the estimation is based on a routine laboratory test, which is available in most hospitals.

## 1. Introduction

Cholangiocarcinoma (CCA) is a relatively rare biliary tract malignancy, accounting for 15–20% of primary liver cancers [1]. High incidence is reported in Thailand, Korea, Japan, and China [1]. CCA is generally diagnosed in the advanced stage due to its asymptomatic nature and non-specific diagnostic tools for early-stage disease [1]. The gold standard of CCA diagnosis is magnetic resonance cholangiopancreatography (MRCP), which has moderate-to-high accuracy and requires further cytological/histological confirmation [2]. Due to the advanced stage of CCA and associated risks [1], the invasive procedure, tumor biopsy, is limited to only some patients.

Several cancer biomarkers have been developed for intrahepatic cholangiocarcinoma (iCCA) diagnosis, including biliary alkaline phosphatase, carbohydrate antigen 19-9 (CA19-9), cholangiocarcinoma-associated carbohydrate antigen (CCA-CA), carcinoembryonic antigen (CEA), cytokeratine 19 fragment 21-1 (CYFRA 21-1), human telomerase reverse transcriptase (hTERT), interleukin-6 (IL6), matrix metalloproteinase-7 (MMP7), and mucin 5AC (MUC5AC) [3]. Nevertheless, the predictive accuracy of these biomarkers is moderate-to-high due to the heterogeneity of iCCA. CA19-9 is the only liquid biopsy tool for clinical use, but the sensitivity and specificity are low [3]. Changes in pro-inflammatory and anti-inflammatory cytokine levels, i.e., interleukin 2 (IL2), IL4, IL6, IL10, IL17A, interferon-gamma (IFN-ꝩ), and tumor necrosis factor-alpha (TNF-α), are observed in iCCA patients [4,5], which could serve as potential iCCA diagnostic markers. In addition, peripheral blood index, e.g., neutrophil-to-lymphocyte ratio (NLR), monocyte-to-lymphocyte ratio (MLR), lymphocyte-to-monocyte ratio (LMR), platelet-to-lymphocyte ratio (PLR), and systemic-immune inflammation index (SIII), are considered potential prognostic factors for assessing clinical outcomes in iCCA patients [6,7,8]. The diagnostic utility of these markers is, however, limited. This study investigated the potential utility of these markers for iCCA diagnosis.

## 2. Methods

### 2.1. Healthy Subjects

The study (*n* = 48) was conducted at Thammasat University, Pathumthani, Thailand, as a part of an opened-labeled randomized placebo-controlled design phase 1 clinical trial (Thai Clinical Trials Registry No. TCTR20201020001). The protocol for the investigation was approved by the Thammasat University Ethics Committee (MTU-EC-OO-3075/61) [4].

### 2.2. iCCA Patients

The study (*n* = 47) was conducted at Sakhon Na-Kon Hospital, Sakhon Na-Kon, Thailand, as a part of a single-center, open-label, randomized, controlled phase 2A trial (registration No. TCTR20210129007) [5]. The study protocol was approved by the Ethics Committee of Sakhon Na-Kon Hospital (No. 04/2564). The research complied with good clinical practice (GCP) guidelines and the Helsinki Declaration.

### 2.3. Inclusion Criteria

Patients aged 18 years or older who had advanced-stage iCCA and had declined standard chemotherapy were enrolled in the study. The inclusion criteria were (i) confirmation of unresectable or metastatic iCCA using ultrasonography, contrast-enhanced computed tomography (CT), or magnetic resonance cholangiopancreatography (MRCP), with serum levels of carcinoembryonic antigen (CEA) greater than 5.2 ng/mL and serum carbohydrate antigen 19-9 (CA19-9) greater than 129 U/mL; (ii) the presence of at least one measurable tumor lesion (with the longest diameter being at least 20 mm) according to the Response Evaluation Criteria in Solid Tumors (RECIST: version 1.1); (iii) an Eastern Cooperative Oncology Group (ECOG) status ranging from 0 to 2; (iv) no prior chemotherapy or radiotherapy and the absence of cardiac abnormalities or abnormal electrocardiograms (ECGs); (v) adequate bone marrow and organ functions; (vi) no abnormal blood clotting or bleeding and no use of anticoagulants or antiplatelets; (vii) practical communication skills; (viii) and willingness to provide informed consent for study participation.

### 2.4. Exclusion Criteria

Exclusion criteria included: (i) pregnancy or lactation; (ii) hypersensitivity or idiosyncratic reactions to any medication or herbal product; (iii) current or past diagnosis of other cancers within five years; (iv) Crohn’s disease or ulcerative colitis; (v) conditions associated with immune deficiency such as HIV, HTLV, tuberculosis, or systemic lupus erythematosus; or (vi) participation in any other study within the previous three months.

### 2.5. Laboratory Procedures

Procedures for the isolation of peripheral mononuclear cells (PBMCs), RNA extraction, cDNA synthesis, mRNA expression quantification of IL6 and TNF-α, and the determination of cytokine levels are described in the Appendix A.

### 2.6. Statistical Analysis

Statistical analysis methods are described in the Appendix A.

## 3. Results

### 3.1. Baseline Characteristics

Baseline characteristics of healthy subjects and advanced-stage iCCA patients are summarized in Table 1. Flow charts of the exclusion of both groups of subjects are shown in Figure 1 and Figure 2.

Healthy subjects had significantly lower age (22.5 vs. 65.5 years, *p* < 0.001), PT (11.6 vs. 13.49, *p* < 0.001), INR (0.98 vs. 1.15, *p* < 0.001), PTT (26.95 vs. 29.4, *p* = 0.002), BUN (10.2 vs. 13.55, *p* = 0.003), and AST (14 vs. 43, *p* < 0.001) and higher BMI (22.67 vs. 21.87 kg/m^2^, *p* = 0.04) and albumin (4.2 vs. 3.8, *p* < 0.001) than the iCCA patients. In addition, they had significantly lower NLR (1.45 vs. 3.22, *p* < 0.001), PLR (112.36 vs. 161.85, *p* < 0.001), and SIII (×10^9^/L) (370.36 vs. 918.55, *p* < 0.001) than the iCCA group (Table 1). LMR was significantly higher in the healthy group (10.10 vs. 3.8, *p* < 0.001). No significant differences in direct bilirubin, total bilirubin, and total protein were found.

Significantly lower levels of IL2 (0 vs. 8.5, *p* < 0.01), IL4 (0.02 vs. 4.25, *p* < 0.001), IL10 (1.75 vs. 5.91, *p* < 0.01), IL17A (0 vs. 75.55, *p* < 0.01), IFN-ꝩ (0 vs. 9.85, *p* < 0.001), and TNF-α (0 vs. 6.66, *p* < 0.001) pg/mL were found in the iCCA group compared with the healthy group. IL6 levels, on the other hand, were significantly higher in the iCCA group (20.91 vs. 8.45, *p* < 0.001).

### 3.2. Interplay between Peripheral Blood Cytokines

The levels of different cytokines in the healthy group were significantly correlated, i.e., IFN-γ vs. TNF-α (R^2^ = 0.478, *p* = 0.002), IFN-γ vs. IL2 (R^2^ = −0.462, *p* = 0.003), IFN-γ vs. IL17A (R^2^ = 0.661, *p* < 0.001), IL2 vs. IL4 (R^2^ = 0.507, *p* < 0.001), IL2 vs. IL10 (R^2^ = 0.591, *p* < 0.001), IL2 vs. IL17A (R^2^ = -0.498, *p* = 0.001), IL4 vs. IL10 (R^2^ = 0.805, *p* < 0.001), and IL4 vs. IL17A (R^2^ = 0.426, *p* = 0.006) (Appendix A). For the iCCA group, significant correlations were found with IFN-γ vs. IL2 (R^2^ = +0.542, *p* = 0.002), TNF-α vs. IL4 (R^2^ = +0.525, *p* = 0.003), and IL6 vs. IL10 (R^2^ = +0.404, *p* = 0.027) (Appendix A).

### 3.3. ROC Analysis

For LMR, the AUC (Figure 3), sensitivity, specificity, NPV, PPV, AI, and YI were 0.91 (0.85–0.97), 0.91, 0.85, 0.89, 0.87, 0.88, and 0.76, respectively. The cut-off value was 7.2. A suspected case with an LMR of <7.2 was at risk of iCCA development [OR: 58.08 (17.48–378.56), *p* < 0.001]. The AUC, sensitivity, specificity, NPV, PPV, AI, and YI for NLR were 0.882 (0.806–0.958), 0.9, 0.8, 0.8, 0.9, 0.85, and 0.7, respectively. The corresponding values for PLR were 0.641 (0.519–0.763), 0.74, 0.71, 0.64, 0.8, 0.72, and 0.45, respectively. The NLR and PLR cut-off values in patients at risk of iCCA development were an NLR of >1.95 and a PLR of 124.

For serum cytokine IL6, the AUC (Figure 4), sensitivity, specificity, NPV, PPV, AI, and YI were 0.81 (0.68–0.93), 0.7, 0.91, 0.82, 0.84, 0.82, and 0.61, respectively. The cut-off value for IL6 was 11.63. Patients with an IL6 of >11.63 pg/mL had a 23-fold increased risk of iCCA [OR: 23.33 (6.42–84.83), *p* < 0.001] compared with those with a level of ≤11.635 pg/mL. The AUC, sensitivity, specificity, NPV, PPV, AI, and YI cut-off values and OR values for IL2, IL4, IL10, IL17A, TNF-α, and IFN-γ are shown in Table 2.

### 3.4. Univariate Analysis of Differential Peripheral Blood Indices and Serum Cytokines

For peripheral blood indices, the univariate analysis results were as follows: NLR (OR: 7.81 (2.95–20.70, *p* < 0.001), LMR (OR: 0.57 (0.45–0.72), *p* < 0.001), PLR (OR 1.01 (1.003–1.017), *p* = 0.006), and SIII (OR 1.004 (1.002–1.006), *p* < 0.001). For serum cytokines, the results were as follows: IL2 (OR:0.528 (0.391–0.712), *p* < 0.001), IL4 (OR: 0.386 (0.239–0.623), *p* < 0.001), IL6 (OR: 1.20 (1.051–1.375), *p* = 0.007), and IL17A (OR: 0.76 (0.61–0.94), *p* = 0.013).

### 3.5. Regression Analysis

Based on multicollinearity analysis, LMR (VIF = 1.6) and PLR (VIF = 2.4) were significant peripheral blood indices (VIF = 1.6). Based on multi-logistic regression analysis, LMR (*p* < 0.0175) was the only significant index. AIC and McFadden’s scores for the training and validation sets were 43.65 and 0.51 and 30.24 and 0.293, respectively.

The IL6 (VIF = 1.56) was a significant peripheral cytokine following multicollinearity analysis and was selected for further logistic regression analysis. AIC and McFadden’s scores for the training and validation sets were 49.34 and 0.33, and 19.46 and 0.44, respectively.

McFadden’s scores for the training and validation sets for LMR and IL6 are in the acceptable ranges (0.2–0.5), indicating the reliability of the model analysis.

### 3.6. Probability Risk of iCCA

The probability risks of iCCA for LMR (Appendix A) and IL6 (Appendix A) were estimated using multi-logistic regression analysis based on probability calculation. The probability risks of LMR and IL6 were classified into four levels, i.e., low (0–25%), intermediate (>25–50%), high (>50–75%), and very high (>75–100%).

Patients with LMR scores of 1–3.38, >3.38–5.77, >5.77–71.8, and >7.18 were classified as very high risk, high risk, intermediate risk, and low risk, respectively (Table 3). Those with IL6 levels of >21.83, 15.94–≤21.33, 8.88–≤15.94, and <8.8 pg/mL were classified as very high risk, high risk, intermediate risk, and low risk, respectively (Table 3).

## 4. Discussion

### 4.1. Peripheral Blood Indices

Baseline NLR, LMR, PLR, and SIII levels in the healthy [9,10,11] and iCCA groups [8,11,12,13] were comparable to the previously reported studies. PLR and SIII were, however, relatively higher [14]. Notably, a clinical study showed reduced PLR and SIII values in early-stage iCCA but not in advanced-stage iCCA patients. These observations may suggest a possible association between increased PLR and SIII levels and disease progression.

### 4.2. Demographic Data

Although the kidney-function and liver-function test values in patients with advanced-stage iCCA were higher than those of healthy volunteers, these parameters were within their normal ranges. The higher kidney-function and liver-function test values in patients may have been due to aging processes since the patients were elderly while the healthy volunteers were young adults.

### 4.3. Peripheral Cytokines

IL6 levels in healthy individuals were lower than the previously reported levels. The finding of higher levels of IL6 in iCCA patients compared with healthy subjects was in agreement with a previous study [15,16]. It is noted, however, that a relatively lower level of IL6 (60-fold) was observed in iCCA patients in this study compared with those previously reported (20 vs. 1200 pg/mL) [17]. Activation of IL6 through STAT3 (signal transducer and activator of transcription 3) in cancer-associated fibroblasts (CAF) has been shown to enhance the resistance of CCA to gemcitabine [17]. Patients with low IL6 levels are, thus, likely to have favorable clinical outcomes. Serum IL6 is also correlated with tumor burden [18] and is suggested as a tool for monitoring treatment outcomes.

IFN-γ and TNF-α levels in iCCA patients were lower than in a previous study [16]. The lower IFN-γ level found in the current study could be due to the AJCC staging (stage III) of most patients, who were more elderly than those in a previous report. Age-associated reduction of IFN-γ levels has also been reported [19]. IFN-γ and TNF-α levels in the iCCA group were lower than in healthy individuals. In *Opisthorchis viverrini* (OV)-induced CCA patients, on the other hand, IFN-γ was relatively high compared with healthy individuals [16]. Notably, low IFN-γ and TNF-α levels have been correlated with high IL6 levels. IFN-γ and TNF-α regulate tumor development and progression in several cancers, including iCCA. Both synergistically upregulate the PD-L1 (programmed death ligand I) expression and impair T-cell mediated anti-tumor immune responses [20]. PD-L1 correlates with the adaptive immunity of CD8+ infiltration in iCCA patients, in which IFN-γ activates in T lymphocytes [20]. Advanced-stage iCCA patients with decreased TNF-α and IFN-γ levels would be expected to have favorable clinical outcomes [20]. Lower IFN-γ levels following nivolumab and gemcitabine therapy resulted in prolonged progression-free survival [21]. Short overall survival of patients with positive PD-L1 was associated with advanced-stage iCCA [21]. Although iCCA patients in this study were in the advanced stage, those with low IFN-γ levels would have a favorable disease prognosis.

IL2 levels in iCCA patients in this study were lower than previously reported [16], which could be due to the age-related decrease in IL2 production [22]. IL2 regulates the production of IFN-γ; thus, the reduction of the IL2 level parallels the decline of IFN-γ output [22]. IL2 serves as an immunomodulator by promoting the cytotoxicity of CD8+ and NK cells [22]. Depletion of IL2 production in iCCA patients could compromise its anti-tumor effect by attenuating the anti-tumor functions of CD8+ and NK cells, resulting in tumor progression. *OV-associated* CCA patients had higher serum IL2 levels than healthy subjects, while serum IL2 levels in cases of PSC were lower [16]. Interestingly, IL2RB expressions in subtype-II (I2) iCCA patients (accounting for 9% of total iCCA types) were higher than non-I2 subtypes [23]. These data support the observation that most iCCA patients are classified as non-I2 subtypes.

IL4 levels in iCCA patients were lower than in patients with *OV-associated* iCCA [16]. Lowering serum IL4 levels is associated with aging [19]. Therefore, low IL4 levels in iCCA patients are associated with age [16]. IL4 plays a crucial role in tumor-promoting survival and growth. IL-4R upregulation is correlated with lower survival in CCA patients [24]. As shown in this study, it is suggested that a depletion in serum IL4 levels in iCCA patients would result in favorable clinical outcomes [24].

IL10 levels in iCCA patients were lower than in a previous study [16]. IL10 is an immunosuppressive cytokine secreted by regulatory B cells, facilitating CCA evasion by attenuating effector T-cell functions [25]. It enhances the polarization of the M2 macrophage via M2 polarized-associated macrophage (M2-TAM) [26], promoting the malignancy of iCCA. IL10 levels are correlated with PD-L1 expression [27]. The higher the extent of the increase in IL10 levels, the higher the PD-L1 expression. Patients with high IL10 levels had shorter overall survival times [28]. Similarly, upregulation of IL10 expression was reported in *OV*-induced CCA patients [16]. Therefore, the iCCA group in the current study would have a favorable overall survival.

Plasma IL17A levels were significantly elevated in CCA patients compared with healthy individuals or patients with liver fluke infections [16]. In this study, however, the IL17A levels in healthy subjects were higher than those with advanced-stage iCCA. These conflicting results may be due to the difference in the type of biological fluids used (serum and plasma) as well as aged-associated cytokine production and stage of the disease [29]. IL17A is an essential signaling pathway in CAFs [30]. CAF-secreted IL17A promotes chemoresistance by activating NF-kB/STAT3 (nuclear factor kappa B-dependent activation of signal transducer and activator of transcription 3) pathways [30]. Significant positive correlations were found between IL17A and PD-L1 expression [31], suggesting that IL17A upregulates PD-L1 expression (activation of HIF1-α; hypoxia-inducible factor-1-alpha) [31], which results in the exhaustion of cytotoxic CD8+ T cells. IL17A augments IL6 production by activating tumor-intrinsic STAT3 [22], resulting in tumor growth and invasion. Accordingly, the depletion of IL17A levels may provide favorable overall survival for iCCA patients.

### 4.4. LMR and IL6 Are Potential Diagnostic Markers for iCCA

LMR and IL6 are identified as significant diagnostic markers of iCCA. LMR’s AUC (0.91) is higher than the cut-off value (0.71), indicating its high predictability performance. Its high sensitivity (91%) and AI (88%) reflect high efficiency in diagnosing iCCA. The sensitivity and specificity of LMR are comparable to the gold standard for iCCA diagnosis (imaging methods with biopsy). LMR estimated from a routine complete blood count (CBC) analysis could be used as a tool for iCCA diagnosis in most hospital settings and is recommended as a preliminary screening test for the diagnosis of CCA before confirmatory testing. The high sensitivity of LMR has been demonstrated when compared with available diagnostic markers, except P53 [3]. Five of the sixteen biomarkers, e.g., CCA-CA, hTERT mRNA, M2-PK, periostin, and IL6, showed sensitivity higher than 80%, with specificity ranging from 26% to 98% [3]. Although these biomarkers provide adequate diagnostic accuracy for CCA, most hospitals’ clinical uses are limited. CEA and CA19-9 are commonly applied to diagnose CCA in most hospitals, although their sensitivity and specificity need to be improved.

In addition to LMR, NLR was shown to be a potential CCA diagnostic marker, with a sensitivity of 70.9% and specificity of 84% [11]. Precision and recall of NLR in the previous study were, however, lower than LMR, indicating that LMR is more accurate for diagnosing iCCA than NLR. In addition, the validation of its applicability is limited. Sex, age, biliary duct stones, and status of portal hypertension were independent prognostic indicators for discriminating combined hepatocellular carcinoma (cHCC) and iCCA [32]. A C-index (Harrell’s concordance index) of 0.796 with a sensitivity of 78% and specificity of 75% for a training set suggests high performance of model accuracy [33]. The sensitivity of the validation set was 68% [32].

A significant decrease in the sensitivity of the validation set by >10% of the training set (training vs. validation sets: 78% vs. 68%) indicates the non-generalizability of the nomogram to other populations due to the overfitting of the model. The current study demonstrated the probability risk of iCCA based on the LMR parameter. This classification based on the risk scores (i.e., very high risk, high risk, intermediate risk, and low risk) would facilitate and encourage the utility and applicability (a preliminary screening test for iCCA diagnosis) of LMR for clinicians in clinical practice, especially those in non-tertiary hospitals. Due to the high prevalence of iCCA in developing countries, high-cost tests may impede the availability of novel modality diagnostic markers. The cost of a CA19-9 assay is about nine times that of a CBC test (LMR). LMR would be more cost-effective than the CA19-9 assay for iCCA screening in endemic areas.

The interpretation and conclusion of the study are limited by the relatively low number of participants in each group and the absence of external validation. In addition, the diagnosis of iCCA was not confirmed by histopathological examination of tumor biopsy. This is due to the fact that the patients included in the study were those who refused chemotherapy as well as biopsy. Further research in this area is essential to elucidate the mechanisms underlying these effects and their implications for overall patient health and well-being. In addition, the information on baseline LMR in the common misdiagnosis of diseases by MRI, e.g., PSC and HCC, was not included in the analysis. The LMR values in HCC and iCCA have been shown to be overlapped [33]. A large clinical trial in patients with PSC and HCC is required to confirm the applicability of LMR as a diagnostic screening tool. Chen and colleagues introduced a nomogram that utilizes inflammatory indices to distinguish between iCCA and HCC. The nomogram has been validated using the C-index, which measures the accuracy of the diagnostic tool [34]. Thus, we proposed utilizing this LMR as a preliminary test, followed by a nomogram, to distinguish between intrahepatic cholangiocarcinoma (iCCA) and hepatocellular carcinoma (HCC). In addition, a further retrospective cohort study in the early stage of iCCA is suggested, since there are no effective biomarkers to detect patients with early-stage iCCA. As a result of early detection and proper management, patients will have a better prognosis.

Pro-inflammatory cytokine levels in iCCA may be affected by comedication, specifically antihypertensive drugs like losartan and enalapril. Individuals with hypertension display elevated levels of pro-inflammatory cytokines, including IL-1β, IL-6, IL-8, IL-17, IL-23, TGF-β, and TNF-α. For example, IL-6 impacts the body’s reactions to angiotensin II infusion, even in patients with normal blood pressure, but IL-17 plays a vital role in the development of hypertension. TNF-α may contribute to hypertension by increasing the synthesis of angiotensin-converting enzyme (ACE). Antihypertensive drugs, such as ACE inhibitors (ACEIs), angiotensin receptor blockers (ARBs), dihydropyridine-calcium channel blockers (DHP-CCBs), thiazide-like diuretics, and beta-blockers, have various effects on immune-cell activity. ACE inhibitors hinder the production of angiotensin II, which leads to the widening of blood vessels and a decrease in blood pressure. On the other hand, ARBs prevent angiotensin II from binding to its receptors, affecting immune-cell activation. DHP-CCBs influence calcium influx in immune cells, thereby regulating signaling and activation, whereas beta-blockers and thiazide-like diuretics have variable effects on immune-cell function [35]. Nevertheless, the individuals enrolled in this research represent fewer than 25% of the patients taking additional medications (comedication). This study suggests that antihypertensive medicines are unlikely to impact pro-inflammatory cytokine levels.

In conclusion, the current study is the first to apply LMR for iCCA diagnosis as a preliminary screening. It estimates disease risk based on four classification levels based on the risk probabilities. The findings might assist clinicians (including general practitioners) in detecting high-risk iCCA cases, leading to further investigation and early intervention. Due to the accessibility and affordability of a CBC test, this diagnostic marker could be applied in non-tertiary hospitals, particularly those in developing countries (endemic areas of iCCA). The limitation of this study, however, is noted. The diagnosis of iCCA was based on clinical signs and symptoms in conjunction with radiological examinations. All patients refused to receive chemotherapy; therefore, tissue biopsies and histological examinations were not performed in all cases to confirm the diagnosis of iCCA.

## Figures and Tables

**Figure 1 biomedicines-12-00844-f001:**
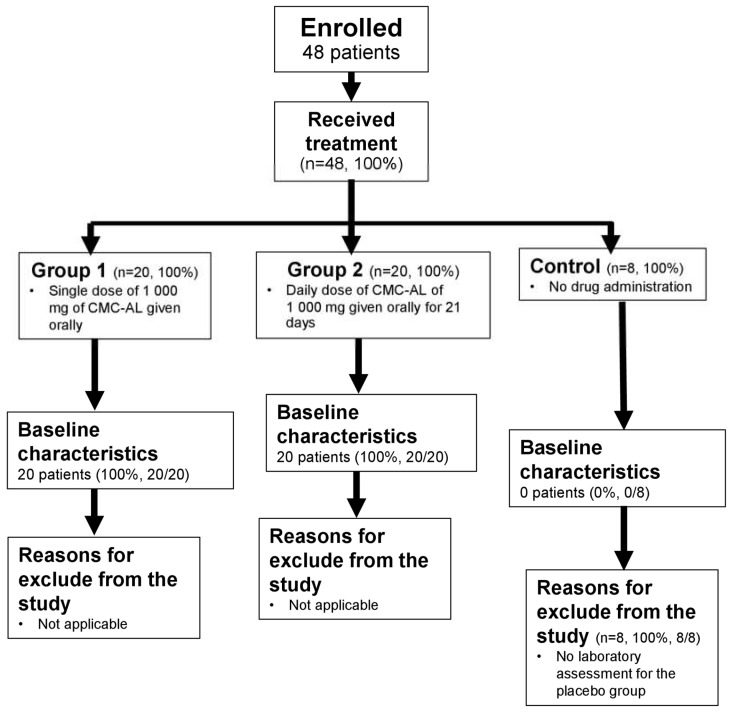
A flow chart of reasons for exclusion from the study.

**Figure 2 biomedicines-12-00844-f002:**
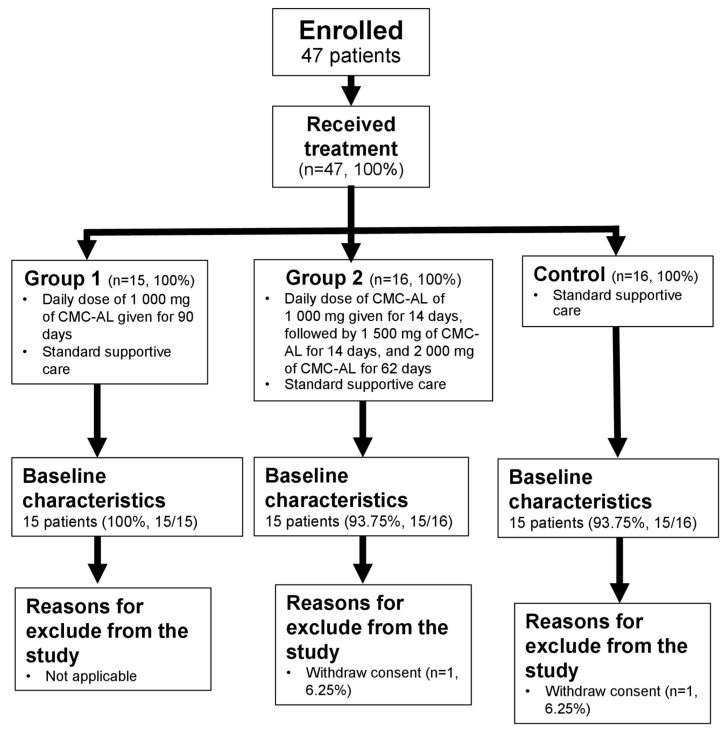
A flow chart of reasons for exclusion of patients with advanced-stage iCCA from the study.

**Figure 3 biomedicines-12-00844-f003:**
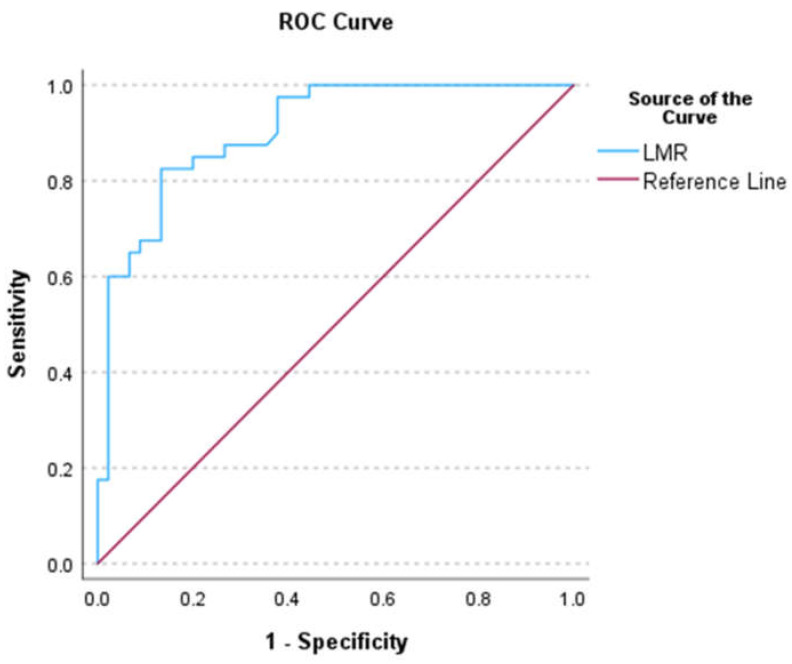
ROC of LMR.

**Figure 4 biomedicines-12-00844-f004:**
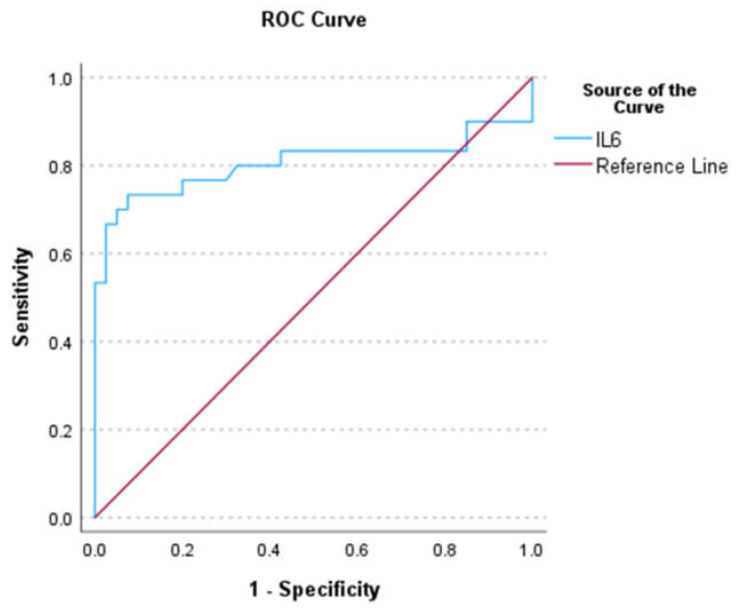
ROC of IL6.

**Table 1 biomedicines-12-00844-t001:** Baseline characteristics of the patients included in the study. Data are presented as median (min–max) or mean ± SD.

Characteristics	Healthy Individuals (*n* = 40)	Patients (*n* = 45)
Gender	Male: 20; Female: 20	Male: 24; Female: 21
Age (years)	22.5 (19–43)	65.5 (45–84)
BMI	22.88 (18.47–26.78)	22.02 (13.03–28.03)
Body temperature (°C)	36.5 (35.7–37.6)	36.6 (36–37.2)
Concomitant medication		
		Antihypertensive (enalapril, amlodipine, spironolactone, furosemide, lercannidine, losartan, and doxazosin), Dyslipidemia (simvastatin), Antibiotics (cefixime, ciproflex, and metronidazole), Antidiabetics (metformin), Analgesics (tramadol, paracetamol, and morphine), Antihistamine (dimenhydrinate), Others (morphine, omeprazole, domperidone/Gabapentin/Amitryptiline, and senoside)
NO	-	35
PT (seconds)	11.6 (±0.72) *	13.49 (±1.66) *
INR	0.98 (±0.06) *	1.15 (±0.15) *
PTT (seconds)	26.95(22.3–31.8)	29.4 (8.7–35.9)
Creatinine (mg/dL)	0.86 (0.5–1.11)	0.89 (0.5–1.98)
BUN (mg/dL)	10.2 (5.4–19.7)	13.55 (0.5–31)
BUN/Cr ratios	13.66 (7.01–31.77)	15.04 (1–30.78)
AST (IU/L)	14 (7–32)	37.5 (13–98)
ALT (IU/L)	30 (16–68)	28 (11–98)
Direct bilirubin (mg/dL)	0.2 (0.1–0.3)	0.2 (0.1–3.5)
Total bilirubin (mg/dL)	0.6 (0.2–1.5)	0.65 (0.2–8)
Total protein (g/dL)	7.9 (7.2–8.8) *	8.06 (±0.71) *
Albumin (g/dL)	4.2 (3.4–4.9)	3.28 (3.04–3.52)
CEA (ng/mL)	-	6.1 (0.5–1111.30)
CA19–9 (ng/mL)	-	116 (0.5–140,000)
NLR	1.55 (±0.45) *	3.22 (0.84–15.16)
LMR	10.10 (4.48–18.26)	3.8 (1–14.90)
PLR	117.87 (±33.43) *	161.85 (10.12–623.79)
SIII (×10^9^/L)	370.36 (189.37–691.11)	918.86 (42.5–6354.83)
IFN-ꝩ (pg/mL)	9.85 (4.4–20.48)	0 (0–3.44)
TNF-α (pg/mL)	6.82 (±1.38) *	0 (0–4.54)
IL2 (pg/mL)	8.49 (0–25.56)	0 (0–5.38)
IL4 (pg/mL)	4.27 (0–14.25)	0.02 (0–2.02)
IL6 (pg/mL)	8.45 (5.08–19.39)	20.91 (3–119.51)
IL10 (pg/mL)	5.91 (2.06–8.63)	1.75 (0–7.15)
IL17A (pg/mL)	75.55 (27.61–214.62)	0 (0–33.37)

BMI: body mass index; IL2: interleukin-2; IL4: interleukin-4; IL6: interleukin-6; IL10: interleukin-10; IL17A: interleukin-17A; LMR: lymphocyte-to-monocyte ratio; NLR: neutrophil-to-lymphocyte ratio; PLR: platelet-to-lymphocyte ratio; SIII: systemic immune–inflammation index; *: normal distribution.

**Table 2 biomedicines-12-00844-t002:** The AUC, sensitivity, specificity, NPV, PPV, AI, and YI cut-off values and OR for IL2, IL4, IL10, IL17A, TNF-α, and IFN-γ.

Cytokines	IL2	IL4	IL10	IL17A	TNF-α	IFN-γ
AUC [median (min–max)]	0.91 (0.84–0.98)	0.84 (0.75–0.93)	0.93 (0.86–0.99)	0.99 (0.99–1.0)	0.99 (0.99–1.0)	0.99 (0.99–1.0)
Sensitivity	0.82	0.74	0.74	0.97	0.97	1.0
Specificity	0.97	0.97	0.97	0.98	0.98	0.98
PPV	0.96	0.97	0.97	0.97	0.97	0.97
NPV	0.85	0.74	0.75	0.98	0.98	1
AI	0.9	0.84	0.84	0.97	0.97	0.98
YI	0.8	0.71	0.71	0.94	0.94	0.97
Cut-off value for risk of iCCA (pg/mL)	<3.24	<1.89	<4.66	<28.10	<4.47	<2.21
OR ratios [Mean ± 95%CI]	183.7 (24.24–1911, *p* < 0.001)	77.33 (12.21–813.1, *p* < 0.001)	50 (11.92–161.9, *p* < 0.001)	1247 (79.50–12,511)	1247 (79.50–12,511)	∞

**Table 3 biomedicines-12-00844-t003:** Risk-probabilities classification of LMR and IL6.

Risk Level	LMR	IL6 (pg/mL)
Low risk	>7.18	<8.8
Intermediate risk	5.78–7.18	8.8–15.94
High risk	3.38–5.77	15.93–21.83
Very high risk	1–3.38	>21.83

LMR: lymphocyte-to-monocyte ratio; IL6: Interleukin-6.

## Data Availability

The datasets used and/or analyzed during the current study are restricted due to the data potentially identifying or sensitive patients’ information. Contact information: kesaratmu@yahoo.com.

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
