# Peer review of "Interleukin-6 and Lymphocyte-to-Monocyte Ratio Indices Identify Patients with Intrahepatic Cholangiocarcinoma"

_biomedicines, 2024, doi:10.3390/biomedicines12040844_

Round 1
Reviewer 1 Report
Comments and Suggestions for Authors
Dear editors:
It is a great honor and pleasure for me to be invited as the reviewer for this important work entitled “Baseline peripheral blood indices and peripheral serum cytokines identify patients with intrahepatic cholangiocarcinoma”. Teerachat Saeheng and co-authors deeply investigated the clinical application of circulating Interleukin-6 (IL6) and lymphocyte-to-monocyte ratio (LMR) to predict intrahepatic cholangiocarcinoma. This study topic is novel and advanced, attributing to corresponding author Kesara Na Bangchang’s long-term efforts and contributions in this scientific field. I have a few comments concerning this study:
1. The title should be more concise. “Peripheral/ serum…” seems redundant. The terms of blood indices and cytokines should be replaced by Interleukin-6 (IL6) and lymphocyte-to-monocyte ratio to highlight the key player of the study.
2. The figure legends were missing.
3. Table 1: the data presentation indicated all the variables were non-Gaussian distribution. I think it is quite unreasonable.
4. The sample size is too small that the results should be interpreted with caution.
5. The author should provide more bio-demographic characteristics in Table 1.
6. A myriad of inflammatory diseases would raise the levels of cytokines. The authors should provide the levels of CRP, CA-199 and other cancer biomarkers. How did authors differentiate patients with inflammation, iCCA, and other cancers?
7. The study results were scanty that the author should provide more valuable analysis.
8. In fact, IL-4,5,6,10,13 are anti-inflammatory cytokines. IL-2, IFN-ꝩ and TNF-α are pro-inflammatory cytokines. Healthy patients should not exert higher levels of inflammatory cytokines than those with iCCA.
Comments on the Quality of English LanguageExtensive editing of English language is required.
Author Response
SUMMARY OF RESPONSE TO REVIEWERS’ COMMENTS
#Reviewer 1
It is a great honor and pleasure for me to be invited as the reviewer for this important work entitled “Baseline peripheral blood indices and peripheral serum cytokines identify patients with intrahepatic cholangiocarcinoma”. Teerachat Saeheng and co-authors deeply investigated the clinical application of circulating Interleukin-6 (IL6) and lymphocyte-to-monocyte ratio (LMR) to predict intrahepatic cholangiocarcinoma. This study topic is novel and advanced, attributing to corresponding author Kesara Na Bangchang’s long-term efforts and contributions in this scientific field. I have a few comments concerning this study:
- The title should be more concise. “Peripheral/ serum…” seems redundant. The terms of blood indices and cytokines should be replaced by Interleukin-6 (IL6) and lymphocyte-to-monocyte ratio to highlight the key player of the study.
Response: The title has been changed as suggested.
- The figure legends were missing.
Response: The figure legends have been added.
- Table 1: the data presentation indicated all the variables were non-Gaussian distribution. I think it is quite unreasonable.
Response: All variables have been re-analysed as suggested.
- The sample size is too small that the results should be interpreted with caution.
Response: This limitation has been discussed.
- The author should provide more bio-demographic characteristics in Table 1.
Response: The additional bio-demographic data characteristics have been added in Table 1 as follows:
Table 1. Baseline characteristics of the patients included in the study. Data are presented as median (min-max) or mean ±95%CI.
|
Characteristics |
Healthy individuals (n=40) |
Patients (n=45) |
|
Gender |
Male: 20; Female: 20 |
Male: 24; Female: = 21 |
|
Age (years) |
22.5 (19-43) |
65.5 (45-84) |
|
BMI |
22.88 (18.47-26.78) |
22.02 (13.03-28.03) |
|
Body temperature (0C) |
36.5 (35.7-37.6) |
36.6 (36-37.2) |
|
Concomitant medication |
|
|
|
|
- |
10 (Antihypertensive (enalapril, amlodipine, spironolactone, furosemide, lercannidine, losartan, and doxazosin), Dyslipidemia (simvastatin), Antibiotics (cefixime, ciproflex, and metronidazole), Antidiabetics (metformin), Analgesics (tramadol, paracetamol, and morphine), Antihistamine (dimenhydrinate), Others (morphine, omeprazole, domperidone/Gabapentin/Amitryptiline, and senoside) |
|
No |
- |
35 |
|
PT (seconds) |
11.6(11.44-11.86)* |
13.49 (8.3-18.68)* |
|
INR |
0.98 (0.97-1.00)* |
1.15 (0.7-1.59)* |
|
PTT (seconds) |
26.95(22.3-31.8) |
29.4 (8.7-35.9) |
|
Creatinine (mg/dL) |
0.86 (0.5-1.11) |
0.89 (0.5-1.98) |
|
BUN (mg/dL) |
10.2 (5.4-19.7) |
13.55 (0.5-31) |
|
BUN/Cr ratios |
13.66 (7.01-31.77) |
15.04 (1-30.78) |
|
AST (IU/L) |
14 (7-32) |
37.5 (13-98) |
|
ALT (IU/L) |
30 (16-68) |
28 (11-98) |
|
Direct bilirubin (mg/dL) |
0.2 (0.1-0.3) |
0.2 (0.1-3.5) |
|
Total bilirubin (mg/dL) |
0.6 (0.2-1.5) |
0.65 (0.2-8) |
|
Total protein (g/dL) |
7.9 (7.2-8.8)* |
8.06 (7.78-8.34)* |
|
Albumin (g/dL) |
4.2 (3.4-4.9) |
3.28 (3.04-3.52) |
|
CEA (ng/mL) |
- |
6.1 (0.5-1111.30) |
|
CA19-9 (ng/mL) |
- |
116 (0.5-140,000) |
|
NLR |
1.55 (1.50-1.60)* |
3.22 (0.84-15.16) |
|
LMR |
10.10 (4.48-18.26) |
3.8 (1-14.90) |
|
PLR |
117.87 (114.31-121.44)* |
161.85 (10.12-623.79) |
|
SIII (x109/L) |
370.36 (189.37-691.11) |
918.86 (42.5-6354.83) |
|
IFN-ꝩ (pg/mL) |
9.85 (4.4-20.48) |
0 (0-3.44) |
|
TNF-α (pg/mL) |
6.81 (6.66-6.96)* |
0 (0-4.54) |
|
IL2 (pg/mL) |
8.49 (0-25.56) |
0 (0-5.38) |
|
IL4 (pg/mL) |
4.27 (0-14.25) |
0.02 (0-2.02) |
|
IL6 (pg/mL) |
8.45 (5.08-19.39) |
20.91 (3-119.51) |
|
IL10 (pg/mL) |
5.91 (2.06-8.63) |
1.75 (0-7.15) |
|
IL17A (pg/mL) |
75.55 (27.61-214.62) |
0 (0-33.37) |
BMI: body mass index; IL2: interleukin-2; IL4: interleukin-4; IL6: interleukin-6; IL10: interleukin-10; IL17A: interleukin-17A; LMR: lymphocyte-to-monocyte ratio; NLR: neutrophil-to-lymphocyte ratio; PLR: platelet-to-lymphocyte ratio; SII: systemic immune-inflammation index;*:normal distribution
Healthy subjects had significantly lower age (22.5 vs. 65.5 years, p<0.001), PT (11.6 vs 13.49, p<0.001), INR (0.98 vs 1.15, p<0.001), PTT (26.95 vs 29.4, p=0.002), BUN (10.2 vs 13.55, p=0.003), AST (14 vs 43 p<0.001) and higher BMI (22.67 vs. 21.87 kg/m2, p=0.04), albumin (4.2 vs 3.8 p<0.001) than the iCCA patients. In addition, they had significantly lower NLR (1.45 vs. 3.22, p<0.001), PLR (112.36 vs. 161.85, p<0.001), and SIII (x109/L) (370.36 vs. 918.55, p<0.001) than the iCCA group (Table 1). LMR was significantly higher in the healthy group (10.10 vs. 3.8, p<0.001). No significant differences in direct bilirubin, total bilirubin, and total protein were found.
- A myriad of inflammatory diseases would raise the levels of cytokines. The authors should provide the levels of CRP, CA-199 and other cancer biomarkers. How did authors differentiate patients with inflammation, iCCA, and other cancers?
Response: The CA19-9 and CEA levels in patients have been added to the demographic data. The additional sentence has been added in the limitations as follows: Chen and colleagues introduced a nomogram that utilizes inflammatory indices to distinguish between iCCA and HCC. The nomogram has been validated using the C-index, which measures the accuracy of the diagnostic tool.34 Thus, we proposed utilizing this LMR as a preliminary test, followed by a nomogram, to distinguish between intrahepatic cholangiocarcinoma (iCCA) and hepatocellular carcinoma (HCC). In addition, multi-center studies are suggested to evaluate specificity in misdiagnosed diseases, e.g., cancers and inflammatory diseases, which have increased pro-inflammatory cytokines, and early-stage patients. This study aims to develop a screening method to detect patients at risk of developing iCCA based on the probability risk of LMR (CBC laboratory) and IL6. The gold standard “MRCP” is needed. IL6 can increase in patients who are infected with Opisthorchis viverrini (OV) infection, and those patients are at risk of developing iCCA later due to the ingestion of fermented fish (Pla-ra) infected with (OV), particularly in Northeastern Thailand (endemic areas). In addition to OV infection, IL6 levels increase in PSC (inflammation), but they had different clinical presentations with iCCA, e.g., chronic jaundice, significant weight loss, and cachexia. Further multicenter clinical studies, which include patients with PSC, and HCC are needed.
- The study results were scanty and the author should provide more valuable analysis.
Response: The additional analysis has been added as follows:
The AUC, sensitivity, specificity, NPV, PPV, AI and YI, cut-off values and OR values for IL2, IL4, IL10, IL17A, TNF-α, and IFN-ꝩ are shown in Table 2.
Table 2. The AUC, sensitivity, specificity, NPV, PPV, AI and YI, cut-off values and OR for IL2, IL4, IL10, IL17A, TNF-α, and IFN-ꝩ.
|
Cytokines |
IL2 (pg/mL) |
IL4 (pg/mL) |
IL10 (pg/mL) |
IL17A (pg/mL) |
TNF-α (pg/mL) |
IFN-ꝩ (pg/mL) |
|
AUC [median (min-max)] |
0.91 (0.84-0.98) |
0.84 (0.75-0.93) |
0.93 (0.86-0.99) |
0.99 (0.99-1.0) |
0.99 (0.99-1.0) |
0.99 (0.99-1.0) |
|
Sensitivity |
0.82 |
0.74 |
0.74 |
0.97 |
0.97 |
1.0 |
|
Specificity |
0.97 |
0.97 |
0.97 |
0.98 |
0.98 |
0.98 |
|
PPV |
0.96 |
0.97 |
0.97 |
0.97 |
0.97 |
0.97 |
|
NPV |
0.85 |
0.74 |
0.75 |
0.98 |
0.98 |
1 |
|
AI |
0.9 |
0.84 |
0.84 |
0.97 |
0.97 |
0.98 |
|
YI |
0.8 |
0.71 |
0.71 |
0.94 |
0.94 |
0.97 |
|
Cut-off value for risk of iCCA |
<3.24 |
<1.89 |
<4.66 |
<28.10 |
<4.47 |
<2.21 |
|
OR ratios [Mean ±95%CI] |
183.7 (24.24-1911, p<0.001) |
77.33 (12.21-813.1, p<0.001) |
50 (11.92-161.9), p<0.001 |
1247 (79.50-12511) |
1247 (79.50-12511) |
∞ |
Univariate analysis of differential peripheral blood indices and serum cytokines
For peripheral blood indices, the univariate analysis suggested that NLR (OR: 7.81 (2.95-20.70, p<0.001), LMR (OR: 0.57 (0.45-0.72), p<0.001), PLR (OR 1.01 (1.003-1.017), p=0.006), and SIII (OR 1.004 (1.002-1.006), p<0.001). For serum cytokines, IL2 (OR:0.528 (0.391-0.712), p<0.001), IL4 (OR: 0.386 (0.239-0.623), p<0.001), IL6 (OR: 1.20 (1.051-1.375), p=0.007), IL17A and (OR: 0.76 (0.61-0.94), p=0.013).
- In fact, IL-4,5,6,10,13 are anti-inflammatory cytokines. IL-2, IFN-ꝩ and TNF-α are pro-inflammatory cytokines. Healthy patients should not exert higher levels of inflammatory cytokines than those with iCCA.
Response: We agree with your comments. However, it may be due to increased immunosuppressive cells in iCCA as described in the tumor microenvironment, e.g., regulatory T cells. In addition, most patients in this study are the elderly who may have impaired immune responses compared to those young adults (https://www.frontiersin.org/articles/10.3389/fragi.2020.602108/full; https://www.ncbi.nlm.nih.gov/pmc/articles/PMC3582124/). Pro-inflammatory cytokines play a role in initiating iCCA development, but the roles of those cytokines may be decreased in the advanced stage of iCCA (https://www.sciencedirect.com/science/article/pii/S0925443917302569)
Comments on the Quality of English Language
Extensive editing of the English language is required
Response: The grammar has been corrected by a native English speaker (Mr. Ethan Vindvamara).
Reviewer 2 Report
Comments and Suggestions for Authors
Here are my thoughts on improving this scientific paper:
1. Provide more details on the patient population. The inclusion/exclusion criteria, disease staging, and demographics should be described more thoroughly. This will allow readers to better evaluate the generalizability of the findings.
2. Expand the discussion of limitations. The authors briefly mention the small sample size, but more attention should be given to confounding factors and sources of bias. How might comorbidities or medications affect the biomarker levels?
3. Include a future directions section. Building off the limitations, suggest next steps for validating LMR as a diagnostic tool. Multi-center studies, evaluating specificity in misdiagnosed diseases, and early stage patients are potential areas to cover.
Comments on the Quality of English Language
Overall, the English is at an appropriate academic level befitting a scientific publication. Medical and statistical terminology is used correctly. Some minor grammatical errors could be improved via careful proofreading.
Author Response
Reviewer2
Comments and Suggestions for Authors
Here are my thoughts on improving this scientific paper:
- Provide more details on the patient population. The inclusion/exclusion criteria, disease staging, and demographics should be described more thoroughly. This will allow readers to evaluate the generalizability of the findings better.
Response: The inclusion/exclusion criteria and additional demographic data have been added as suggested.
Inclusion criteria
Patients aged 18 years or older who had advanced-stage intrahepatic cholangiocarcinoma (iCCA) and had declined standard chemotherapy were enrolled in the study. The inclusion criteria were (i) confirmation of unresectable or metastatic iCCA using ultrasonography, contrast-enhanced computed tomography (CT), or magnetic resonance cholangiopancreatography (MRCP), with serum levels of carcinoembryonic antigen (CEA) greater than 5.2 ng/ml and serum carbohydrate antigen 19-9 (CA19-9) greater than 129 U/ml; (ii) the presence of at least one measurable tumor lesion (with the longest diameter of at least 20 mm) according to Response Evaluation Criteria in Solid Tumors (RECIST: version 1.1); (iii) Eastern Cooperative Oncology Group (ECOG) status ranging from 0 to 2; (iv) no prior chemotherapy or radiotherapy; absence of cardiac abnormalities or abnormal electrocardiograms (ECGs); (v) adequate bone marrow and organ functions; (vi) no abnormal blood clotting or bleeding; (vii) no use of anticoagulants or antiplatelets; practical communication skills; (viii) and willingness to provide informed consent for study participation.
Exclusion criteria
The exclusion criteria included: (i) pregnancy or lactation; (ii) hypersensitivity or idiosyncratic reactions to any medication or herbal product; (iii) current or past diagnosis of other cancers within five years; (iv) Crohn’s disease or ulcerative colitis; (v) conditions associated with immune deficiency such as HIV, HTLV, tuberculosis, or systemic lupus erythematosus; or (vi) participation in any other study within the previous three months
- Expand the discussion of limitations. The authors briefly mention the small sample size, but more attention should be given to confounding factors and sources of bias. How might comorbidities or medications affect the biomarker levels?
Response: Pro-inflammatory cytokine levels in iCCA may be affected by comedication, precisely antihypertensive drugs like losartan and enalapril. Individuals with hypertension display elevated levels of pro-inflammatory cytokines, including IL-1β, IL-6, IL-8, IL-17, IL-23, TGF-β, and TNF-α. For example, IL-6 impacts the body's reactions to angiotensin II infusion, even in patients with normal blood pressure, but IL-17 plays a vital role in the development of hypertension. TNF-α may contribute to hypertension by increasing the angiotensin-converting enzyme (ACE) synthesis. Antihypertensive drugs, such as ACE inhibitors (ACEIs), angiotensin receptor blockers (ARBs), dihydropyridine-calcium channel blockers (DHP-CCBs), thiazide-like diuretics, and beta-blockers, have various effects on immune cell activity. ACE inhibitors hinder the production of angiotensin II, which leads to the widening of blood vessels and a decrease in blood pressure. On the other hand, ARBs prevent angiotensin II from binding to its receptors, affecting immune cells' activation. DHP-CCBs influence calcium influx in immune cells, thereby regulating signaling and activation, whereas beta-blockers and thiazide-like diuretics have variable effects on immune cell function.35 Nevertheless, the individuals enrolled in the research represent fewer than 25% of the patients taking additional medications (comedication). The study suggests that antihypertensive medicines are unlikely to impact pro-inflammatory cytokine levels.
- Include a future directions section. Building off the limitations, suggest next steps for validating LMR as a diagnostic tool. Multi-center studies, evaluating specificity in misdiagnosed diseases, and early stage patients are potential areas to cover.
Response: The additional sentence has been added as follows: In addition, multi-center studies, evaluating specificity in misdiagnosed diseases e.g., cancers and inflammatory diseases, which have increasing pro-inflammatory cytokines, and early-stage patients are suggested.
Comments on the Quality of English Language
Overall, the English is at an appropriate academic level befitting a scientific publication. Medical and statistical terminology is used correctly. Some minor grammatical errors could be improved via careful proofreading.
Response: The grammar has been corrected as suggested.
Round 2
Reviewer 1 Report
Comments and Suggestions for Authors
The authors work hard to response to reviewers' comments. However, the data expression of normal distribution in the Table 1 (mean ±95%CI) should be corrected as mean ± SD. Most of all, healthy elder individuals are not patients that should not exert higher levels of inflammatory cytokines than those with iCCA. If they are termed as patients, I ask for authors for the clarification on the definition of elder patients.
Comments on the Quality of English LanguageExtensive editing of English language is required.
Author Response
SUMMARY OF RESPONSE TO REVIEWERS’ COMMENTS
#Reviewer 1
Comments and Suggestions for Authors
The authors work hard to response to reviewers' comments. However, the data expression of normal distribution in the Table 1 (mean ±95%CI) should be corrected as mean ± SD. Most of all, healthy elder individuals are not patients that should not exert higher levels of inflammatory cytokines than those with iCCA. If they are termed as patients, I ask for authors for the clarification on the definition of elder patients.
1) However, the data expression of normal distribution in the Table 1 (mean ±95%CI) should be corrected as mean ± SD.
Respond: The data for normal distribution in Table 1 has been changed as suggested.
Table 1. Baseline characteristics of the patients included in the study. Data are presented as median (min-max) or mean ±SD.
|
Characteristics |
Healthy individuals (n=40) |
Patients (n=45) |
|
Gender |
Male: 20; Female: 20 |
Male: 24; Female: = 21 |
|
Age (years) |
22.5 (19-43) |
65.5 (45-84) |
|
BMI |
22.88 (18.47-26.78) |
22.02 (13.03-28.03) |
|
Body temperature (0C) |
36.5 (35.7-37.6) |
36.6 (36-37.2) |
|
Concomitant medication |
|
|
|
|
|
Antihypertensive (enalapril, amlodipine, spironolactone, furosemide, lercannidine, losartan, and doxazosin), Dyslipidemia (simvastatin), Antibiotics (cefixime, ciproflex, and metronidazole), Antidiabetics (metformin), Analgesics (tramadol, paracetamol, and morphine), Antihistamine (dimenhydrinate), Others (morphine, omeprazole, domperidone/Gabapentin/Amitryptiline, and senoside) |
|
NO |
- |
35 |
|
PT (seconds) |
11.6 (±0.72)* |
13.49 (±1.66)* |
|
INR |
0.98 (±0.06)* |
1.15 (±0.15)* |
|
PTT (seconds) |
26.95(22.3-31.8) |
29.4 (8.7-35.9) |
|
Creatinine (mg/dL) |
0.86 (0.5-1.11) |
0.89 (0.5-1.98) |
|
BUN (mg/dL) |
10.2 (5.4-19.7) |
13.55 (0.5-31) |
|
BUN/Cr ratios |
13.66 (7.01-31.77) |
15.04 (1-30.78) |
|
AST (IU/L) |
14 (7-32) |
37.5 (13-98) |
|
ALT (IU/L) |
30 (16-68) |
28 (11-98) |
|
Direct bilirubin (mg/dL) |
0.2 (0.1-0.3) |
0.2 (0.1-3.5) |
|
Total bilirubin (mg/dL) |
0.6 (0.2-1.5) |
0.65 (0.2-8) |
|
Total protein (g/dL) |
7.9 (7.2-8.8)* |
8.06 (±0.71)* |
|
Albumin (g/dL) |
4.2 (3.4-4.9) |
3.28 (3.04-3.52) |
|
CEA (ng/mL) |
- |
6.1 (0.5-1111.30) |
|
CA19-9 (ng/mL) |
- |
116 (0.5-140,000) |
|
NLR |
1.55 (±0.45)* |
3.22 (0.84-15.16) |
|
LMR |
10.10 (4.48-18.26) |
3.8 (1-14.90) |
|
PLR |
117.87 (±33.43)* |
161.85 (10.12-623.79) |
|
SIII (x109/L) |
370.36 (189.37-691.11) |
918.86 (42.5-6354.83) |
|
IFN-ꝩ (pg/mL) |
9.85 (4.4-20.48) |
0 (0-3.44) |
|
TNF-α (pg/mL) |
6.82 (±1.38)* |
0 (0-4.54) |
|
IL2 (pg/mL) |
8.49 (0-25.56) |
0 (0-5.38) |
|
IL4 (pg/mL) |
4.27 (0-14.25) |
0.02 (0-2.02) |
|
IL6 (pg/mL) |
8.45 (5.08-19.39) |
20.91 (3-119.51) |
|
IL10 (pg/mL) |
5.91 (2.06-8.63) |
1.75 (0-7.15) |
|
IL17A (pg/mL) |
75.55 (27.61-214.62) |
0 (0-33.37) |
2) Most of all, healthy elder individuals are not patients that should not exert higher levels of 4inflammatory cytokines than those with iCCA. If they are termed as patients, I ask for authors for the clarification on the definition of elder patients.
Respond: The healthy subjects included in this study are adults (aged 22.5 (19-43) years) and the patients included in the study are elderly (aged 65.5 (45-84) years). According to the definition of the WHO’s age classification, healthy group is classified as young adults or middle-aged adults, while the patient group is classified as old adults (https://www.who.int/news-room/events/detail/2019/06/12/default-calendar/who-guideline-development-group-for-the-updating-of-the-2010-global-recommendations-on-physical-activity-in-youth-adults-and-older-adults).
3) Comments on the Quality of English Language
Extensive editing of the English language is required.
Respond: The grammar has been corrected by a native English speaker (Dr. Ethan Vindvamara).